# The Effect of Deep Cryogenic Treatment on the Electrocatalytic Performance of a Pd@CFs Catalyst for Methanol Oxidation

**DOI:** 10.3390/nano15050338

**Published:** 2025-02-22

**Authors:** Chenxing Wang, Jiahui Mo, Haoting Wang, Jia Liu, Gege He, Xinhai He, Yanyan Song

**Affiliations:** School of Materials Science and Engineering, Xi’an Key Laboratory of Textile Composites, Xi’an Polytechnic University, Xi’an 710048, China; 221311008@stu.xpu.edu.cn (C.W.); 17392623168@163.com (J.M.); 18568962519@139.com (H.W.); liujia09200920@163.com (J.L.); gghe01@163.com (G.H.)

**Keywords:** CFs, Pd, direct methanol fuel cell, deep cryogenic treatment, crystal orientation

## Abstract

To enhance the electrocatalytic performance of a flexible Pd@CFs catalyst for methanol oxidation, deep cryogenic treatment in liquid nitrogen was introduced. The effects of the frequency and time of the deep cryogenic treatment on the surface crystal orientation, microstructure morphology, mechanical performance, and electrocatalytic performance for methanol oxidation were studied. The results showed that when the frequency of the deep cryogenic treatment was 2 times and the deep cryogenic time was 24 h, the electrocatalytic performance of the catalyst was the best. Compared with the catalyst without deep cryogenic treatment, the activity and stability of the catalyst increased by about 33% and 41%, respectively. The activity and stability of the catalyst were about 43.4 times and 6.3 times that of the commercial Pd/C catalyst, respectively. After 500 cycles of CV testing, the performance of the catalyst decay rate was only 3.9%. Compared to the CFs, the tensile strength and the elongation rates of the catalyst increased by 24.6% and 57%, respectively. This is due to deep cryogenic treatment causing Pd grains to rotate from a disordered arrangement to an ordered arrangement, making the metal particles more dispersed and exposing more active sites, ultimately improving the electrocatalytic oxidation ability of methanol. The excellent electrocatalytic efficiency of Pd@CFs-24-2 coupled with its simple and easy preparation method has great potential for promoting the development of DMFCs.

## 1. Introduction

Nowadays, energy shortages and environmental pollution are severe, and there is an urgent need for a substitute renewable energy source. The development of fuel cell technology provides alternative solutions for constructing green energy [1]. Direct methanol fuel cells (DMFCs) have attracted much attention in the field of high-performance catalytic cells due to their many advantages, such as high energy density, environmental friendliness, convenient fuel sources, and low external environmental requirements during operation [2]. Therefore, they have great potential to become a new type of energy [3,4]. At present, research on DMFCs shows that the practical application of DMFCs is limited due to catalysts being prone to poisoning and due to low catalytic activity. Finding a substance with anti-poisoning and high catalytic activity has become a research hotspot [5,6,7].

Pd is a promising electrocatalytic material because of its high electrocatalytic activity and good anti-toxicity, but its application is limited by its high price [8,9]. Due to the catalytic reaction occurring on the surface of the catalyst, it is important to reduce costs by regulating the morphology, structure, and composition of the Pd catalyst and exposing its active sites to the maximum extent [10]. In addition, the catalytic selectivity of different crystal planes of Pd catalysts also varies. Thus, better electrochemical performance can be achieved by regulating the crystal orientation of active Pd [11]. Deep cryogenic treatment refers to cooling a material to an extremely low temperature to achieve optimization of the material’s microstructure and properties. Deep cryogenic treatment can cause lattice shrinkage and density increase in materials, resulting in phase transformation and gradual stabilization of crystal forms, which can make the grain boundaries of materials clearer and straighter [12,13].

Deep cryogenic treatment has many applications in strengthening the mechanical properties of metals, but there are few applications in improving the catalytic performance of materials. In this work, a Pd@CFs catalyst was obtained by loading Pd onto the surface of flexible carbon fiber sheet (CF) using a straightforward impregnation reduction method. The catalyst surface was subjected to a deep cryogenic treatment to change its crystal orientation and improve its electrocatalytic oxidation methanol performance.

## 2. Experimental Section

### 2.1. Preparation of the Pd@CFs Catalyst

The preparation process of the catalyst is shown in Figure 1. The CF was pretreated by immersing it in 5 mol/L HCl and applying ultrasonic treatment for 5 min, followed by rinsing with deionized water and ethanol, drying at room temperature, and reserving. PdCl_2_ with 4% CF mass was weighed in concentrated HCl until completely dissolved. The CF was immersed in the solution, and ultrasonic treatment was carried out for 30 min. During this process, the pH of the solution was adjusted to 10 using 0.5 mol/L NaOH. Then, a reducing agent NaBH_4_ aqueous solution was added to the solution drop by drop during a slow stirring process to obtain the Pd@CFs catalyst. The samples were cryogenically treated in liquid nitrogen (−196 °C) for a certain period (6 h, 12 h, 24 h, 36 h) and then placed at room temperature for 4 h. The abovementioned deep cryogenic treatment was repeated for a certain number of cycles (1, 2, and 3 times). For clarity, samples without deep cryogenic treatment were designated Pd@CFs, while those with different deep cryogenic treatment cycles and times were named Pd@CFs-a-b (a is the deep cryogenic treatment time in hours, and b is the number of different deep cryogenic treatment cycles).

### 2.2. Characterization

The phase composition of the samples was tested using a BRUKER D8 Discoverer X-ray diffractometer (XRD), the microstructure and morphology were characterized using a Quanta-450-EEG field emission scanning electron microscope (SEM), the elemental species were analyzed using an X-MAX50 energy-dispersive spectrometer (EDS), the elemental composition and chemical state were analyzed using an AXIS SUPRA X-ray photoelectron spectrometer (XPS), the tensile properties were tested using a UTM5504 universal testing machine, and the electrocatalytic oxidation methanol performance was tested using a CHI660E electrochemical workstation.

## 3. Results and Discussion

To clarify the influence of cryogenic times on the crystal structure of noble metal components in the Pd@CFs catalyst, XRD diffraction analysis tests were carried out on samples with different cryogenic times. The XRD and texture coefficient analysis of each sample are shown in Figure 2. The XRD pattern of CFs is shown in Figure 2a, from which diffraction peaks appear to be near 2*θ* = 25.92° (PDF No. 58-1638), which conforms to the characteristic diffraction peaks corresponding to the crystal plane of the CFs’ graphite structures (*002*). The XRD of the Pd@CFs and samples with different cryogenic times are shown in Figure 2b, in which the five diffraction peak positions of the Pd@CFs and each sample treated by the deep cryogenic treatment are located at 40.18°, 46.68°, 67.98°, 81.84°, and 86.76°, and the five characteristic peaks are all attributed to the (*111*), (*200*), (*220*), (*311*), and (*222*) crystal planes of Pd (PDF No. 46-1043); the crystal structure corresponds to the face-centered cubic crystal structure of Pd. Among them, the (*111*) plane diffraction peak intensity of the Pd@CFs sample is the highest, followed by the (*200*) plane diffraction peak intensity. After deep cryogenic treatment, the (*111*) plane diffraction peak intensity and half-maximum width of the sample show a gradual increase trend, and the (*200*) and (*220*) plane diffraction peak intensity show a trend of first strengthening and then weakening. Among them, the position and number of characteristic peaks did not change significantly, and the crystal structure of the catalyst sample was still face-centered cubic (*FCC*), indicating that the crystal structure composition of the sample did not change before and after the deep cryogenic treatment. The preferred orientation of Pd grains on the CFs in the (*hkl*) crystal plane is quantitatively characterized by the texture coefficient *TC*_(*hkl*)_, and the calculation formula of the texture coefficient *TC*_(*hkl*)_ is as follows:(1)$$TC_{\left(\mathrm{hkl}\right)}=\frac{I_{\left(hkl\right)}/I_{0(hkl)}}{\left(\sum_{i=1}^{n}I_{\left(hkl\right)}/I_{0(hkl)}\right)/n}$$

In the formula, *I*_(*hkl*)_ is the intensity of the (*hkl*) crystal plane of the sample, *I*_0(*hkl*)_ is the intensity of the corresponding element standard (*hkl*) crystal plane, and n is the number of characteristic peaks of the corresponding element.

The change trend chart of the crystal plane texture coefficient of the sample is shown in Figure 2c, and the texture coefficient value of each crystal plane is shown in Figure 2d. Combining Figure 2c,d, it can be seen that the (*222*) crystal plane texture coefficient of the Pd@CFs is greater than 1, and when combining the XRD pattern of Figure 2b, it can be seen that the (*222*) peak intensity of each sample is low, which is due to the low standard reference value of the (*222*) crystal plane during the texture coefficient calculation process; other crystal planes are less than 1, which indicates that there is no obvious preferred orientation of Pd loaded on CFs. However, the texture coefficients of the samples after the deep cryogenic treatment in (*200*) and (*220*) crystal planes first increased and then decreased, and both of them were greater than 1, which indicated that the deep cryogenic treatment was helpful for the selection of Pd crystal planes [14,15]. Among them, the (*200*) crystal plane texture coefficient of the Pd@CFs-6-2 is 1.132, and the (*220*) crystal plane texture coefficient is 1.118, which are the largest in the cryogenically treated samples, indicating that its (*200*) and (*220*) crystal planes have the best preferential arrangement effect. The (*111*) crystal plane of each sample can be enhanced and widened, as seen in Figure 2b, but its texture coefficients are all less than 1. This is because the (*200*) and (*220*) crystal planes are enhanced greatly at the same time, and the texture coefficient of the (*111*) crystal plane decreases due to the increase in the denominator. The change in crystal plane orientation may be affected by low-temperature impact during deep cryogenic treatment, and the volume shrinkage effect makes it produce large internal stress. The calculation formula of the internal stress is as follows:(2)σ=E·∆T·α

In the formula, *σ* is the internal stress (MPa), E is the elastic modulus (GPa), Δ*T* is the temperature change during the deep cryogenic treatment (K), and α is the coefficient of thermal expansion (1/K). Substitute *E* = 121 GPa, *α* = 3.8∙10^−6^/K, Δ*T* = 77 K–298 K = −211 K into the above equation to obtain σ = −97 MPa. The stress is negative; therefore, the precious metal in the sample is subjected to compressive stress, resulting in lattice distortion and other defects, and is in a thermodynamically unstable state. During the deep cold recovery stage, the grains rotate towards favorable positions, and the degree of crystal plane orientation deepens [16,17]. The increase in crystal planes with the same orientation improves the order of the crystal, and the more ordered structure produces stronger signals in X-ray diffraction. Therefore, the texture coefficient shows a trend of increasing first, and then the decrease in texture coefficient is due to the recovery or recrystallization process with the prolongation of the deep cryogenic treatment time. The grains begin to rearrange, and the orientation tends to be random, which leads to a decrease in the texture coefficient. Considering the residual stress inside the material, the long-term deep cryogenic treatment relaxes the stress, resulting in the destruction of the orientation order and the decrease in the texture coefficient [18]. In addition, with the increase in the deep cryogenic times, the half-maximum width of (*111*), (*220*), and (*220*) characteristic peaks of the samples all widened in different degrees, which indicated that the difference in internal stress during the deep cryogenic treatment led to the difference in grain size at various stages. Generally speaking, the half-maximum width of diffraction peaks gradually widened with the increase in the deep cryogenic times. This meant that the internal stress produced by the deep cryogenic treatment caused the migration of crystal planes and refined grains. Further improved the uniformity of the crystal structure of noble metals [19,20]. To summarize, the deep cryogenic treatment impacts the crystal structure of noble metal components in the sample. It is speculated that the preferred orientation of crystal planes will lead to changes in the electrocatalytic performance of methanol in the catalyst sample after deep cryogenic treatment.

The SEM, EDS, and XPS spectra of the samples are shown in Figure 3. The SEM diagram of CFs at 500 μm magnification is shown in Figure 3a, from which the CFs penetrate each other and have a uniform thickness. The SEM image of a single CF at a magnification of 10 μm is shown in Figure 3b. The surface of the fiber is smooth, and the average diameter of a single fiber is about 7.5 μm by measuring at different positions. The SEM image of the Pd@CFs at a magnification of 500 μm is shown in Figure 3c. It can be seen from the figure that the morphology of CFs has not changed after Pd loading, and it still shows a staggered network hole structure with Pd particles attached to the fiber surface. The SEM diagram of the Pd@CFs at a magnification of 10 μm is shown in Figure 3d, from which there are more agglomerations of Pd. The SEM image of the Pd@CFs-6-1 at a magnification of 500 μm is shown in Figure 3e. It can be seen from the figure that deep cryogenic treatment does not affect the structure of CFs. The SEM image of the Pd@CFs-6-1 at a magnification of 10 μm is shown in Figure 3f. From a single CF, the surface of the fiber becomes rougher, and there are shallow axial gullies on the surface with an average diameter of 7.22 μm. The Pd load is more uniform than before the deep cryogenic treatment, but there is still a small amount of Pd agglomeration. The SEM diagram of the Pd@CFs-6-2 at a magnification of 10 μm is shown in Figure 3g, from which the Pd dispersion on the CFs surface is more uniform, with almost no Pd agglomeration, and the average fiber diameter is 7.18 μm. The SEM diagram of the Pd@CFs-6-3 at a magnification of 10 μm is shown in Figure 3h, from which it is obvious that the amount of Pd has decreased, and the average fiber diameter is 7.15 μm. By comparing the samples before and after the deep cryogenic treatment, the fiber diameter decreases but the variation range is not large, which may be caused by the volume shrinkage of the CFs during the deep cryogenic treatment [21,22]. In addition, comparing the samples of the Pd@CFs in Figure 3d and the Pd@CFs-6-1 in Figure 3f, it can be found that the loading state of precious metal particles on the surface of the samples after one deep cryogenic treatment is more dispersed, which may be due to the reduction in the number of disordered particles stacked together due to grain rotation during the deep cryogenic treatment, and the macroscopic appearance of metal particles being more dispersed. From the SEM images of the Pd@CFs-6-2 in Figure 3g and the Pd@CFs-6-3 in Figure 3h, it can be seen that the gully of CFs is more obvious, and at the same time, it can be observed that the loading amount of noble metal loaded on the surface of CFs is reduced, which may be due to the volume shrinkage effect of noble metal, which makes the bonding between atoms too tight and causes structural change in CFs, which affects the adhesion with the support, resulting in the loss of noble metal.

The EDS energy spectrum of the Pd@CFs is shown in Figure 3i, and the selected area is shown in Figure 3d; its detection proves that Pd is successfully loaded to the CF surface. The total XPS spectrum of the Pd@CFs sample obtained after standard charge correction using the C1s orbital 284.8 eV is shown in Figure 3g. The spectrum shows that C, O, and Pd elements appear on the sample surface, and the 3d characteristic peak double peak of Pd appears in the range of 336 eV to 346 eV, which is due to the splitting of the characteristic peak caused by the coupling between the angular momentum of the 3d orbital electron of Pd and the spin magnetic field. The XPS spectrum of Pd 3d is shown in Figure 3k. Peak separation treatment shows that two valence states of Pd appear on the surface of the sample [23]. The XPS spectrum of Pd 3d is shown in Figure 3k. Peak separation treatment shows that two valence states of Pd appear on the surface of the sample. They are Pd 3d_3/2_ at 340.58 eV and Pd 3d_5/2_ peaks at 335.28 eV, respectively, both corresponding to the Pd^(0)^ valence state. Similarly, Pd 3d_5/2_ at 337.36 eV and Pd 3d_3/2_ peaks at 342.98 eV, both correspond to the Pd^2+^ valence state, which may be due to the oxidation of the Pd surface during preparation [24]. Figure 3i,j,k indicate that Pd in the Pd@CFs samples was successfully reduced and loaded onto the support and mainly existed in the metallic state, with trace amounts in the oxidation state.

A series of maps of electrocatalytic oxidation of methanol performance tests of samples with different cryogenic times are shown in Figure 4. Samples with different cryogenic times in 1 M KOH solution at a scan rate of 50 mV∙s^−1^ ranging from −1~ to 0.2 V (vs. SCE) The cyclic voltammetry (CV) curve obtained by scanning in the potential range is shown in Figure 4a. From the figure, it can be seen that the profile characteristics of Pd@CFs have little change compared with the CV curve of the sample after the deep cryogenic treatment. In the potential range (−1~−0.7 V), it is the hydrogen region, and in the potential region (0~0.2 V), it is the oxide formation corresponding to Pd. In the negative scanning (−0.5~−0.2 V) potential region, it corresponds to the reduction characteristic peak of PdO. The corresponding electrochemically active surface area (EASA) calculated for the reduction peak area of Pd in each sample is shown in Figure 4b. From the diagram, it can be seen that the EASA values of the Pd@CFs, Pd@CFs-6-1, Pd@CFs-6-2, Pd@CFs-6-3, and commercial Pd/C samples are 14.8 m^2^∙g^−1^, 18.2 m^2^∙g^−1^, 23.3 m^2^∙g^−1^, 9.84 m^2^∙g^−1^, and 0.3 m^2^∙g^−1^, respectively. That is, the EASA value of the sample after the deep cryogenic treatment shows a trend of increasing first and then decreasing, among which the EASA value of the Pd@CFs-6-2 after secondary deep cryogenic treatment is the largest, which is about 77.67 times that of commercial Pd/C.

The CV curves of samples treated with different cryogenic times in 1 M KOH and 1 M CH_3_OH with a scan rate of 50 mV∙s^−1^ are shown in Figure 4c. It can be seen from Figure 4c that with the increase in the number of deep cryogenic treatments, the peak current density of the sample in the forward scan in the CV test shows a trend of first increasing and then decreasing. When the sample was subjected to the deep cryogenic treatment twice, the Pd@CFs-6-2 sample obtained had the highest electrocatalytic oxidation activity for CH_3_OH, with a peak current density of 4191 A∙g^−1^, which was 7% higher than that of the sample without the deep cryogenic treatment (3934 A∙g^−1^), which was 34.8 times that of the commercial Pd/C catalyst (120.4 A∙g^−1^). Although the peak current density is not much higher than that before the deep cryogenic treatment, its positive scanning peak voltage shifts negatively, indicating that the deep cryogenic treatment can improve the catalyst activity and increase its reaction rate. The columnar diagram of the peak current density and voltage of each sample is shown in Figure 4d. It can be clearly seen from the figure that the oxidation potential of the sample after the deep cryogenic treatment gradually shifts negatively. And the Pd@CFs-6-1, Pd@CFs-6-2, and Pd@CFs-6-3 are −0.024 V, −0.027 V, and −0.163 V, respectively. Compared with the Pd@CFs sample without the deep cryogenic treatment (0.09 V), they shift at most by 0.253 V. This indicates that the deep cryogenic treatment can not only improve the electrocatalytic oxidation activity of the catalyst for CH_3_OH but also significantly improve the oxidation kinetic performance of the catalyst for CH_3_OH. Among them, the catalyst peak voltage of Pd@CFs-6-2 is the best, but too many cryogenic times will reduce the activity of the catalyst, which may be caused by the reduction in catalytically active species due to the loss of precious metals [25].

The chronoamperometric curve (i-t) of samples treated with different cryogenic times for CH_3_OH electrocatalytic oxidation is shown in Figure 4e. It can be observed that the current density decays significantly at the initial stage of the test. This is due to the gradual accumulation of carbon-containing intermediates during methanol oxidation, and the adsorption of CO on the catalyst surface leads to a rapid decrease in current density. The current density of each sample begins to tend to a steady state around 2000 s. The steady-state current density of each sample at 5000 s is shown in Figure 4f. The steady-state current density of Pd@CFs-6-1 and Pd@CFs-6-2 samples after primary and secondary deep cryogenic treatment is 46.11 A∙g^−1^ and 50.66 A∙g^−1^, both of which are larger than those of the Pd@CFs without deep cryogenic treatment (42.36 A∙g^−1^). It is about 5.31 times that of commercial Pd/C (9.54 A∙g^−1^), which shows that the deep cryogenic treatment can improve the electrocatalytic stability of the sample, and the best effect is when the number of deep cryogenic treatments is two [26,27].

The CV curve of Pd@CFs-6-2 sample in 1 M KOH and 1 M CH_3_OH solution mixed 500 times is shown in Figure 4g. From Figure 4g, the peak current density of 500 CV cycles is a dynamic change process, and the peak current density decreases from 4191 A∙g^−1^ in the first cycle to 3682 A∙g^−1^ in the 500th cycle, with an attenuation rate of 12.14%. The 500 CV cycles of Pd@CFs-6-2 are shown in Figure 4h. From Figure 4h, we can see the peak current density for every 100 additional cycles. From the figure, with the increase in CV scanning times, the peak current density shows a trend of first increasing and then decreasing [28], but its peak current density changes relatively stable without sudden changes, which further proves that the electrochemical stability of Pd@CFs-6-2 samples is good.

In order to explore the influence of the deep cryogenic treatment on the mechanical properties of the CF carrier, the CF carrier was subjected to the same number of the deep cryogenic treatments without loading Pd. For the convenience of description, the CFs’ supports without the deep cryogenic treatment and with different times of the deep cryogenic treatment are named CFs, CFs-6-1, CFs-6-2, and CFs-6-3, and the stress–strain curves of each sample obtained by the tensile test are shown in Figure 4i. It can be observed from the figure that the curve of all samples is relatively smooth in the initial stage of strain. When the strain develops to a certain extent, the slope of the curve increases rapidly, and the samples also break. For CF samples without the deep cryogenic treatment, the tensile strength is 4.03 MPa, and the elongation is 0.7%. After one deep cryogenic treatment, the tensile strength of the CFs-6-1 is 4.13 MPa, and the elongation is 1.4%. After secondary deep cryogenic treatment, the tensile strength of the CFs-6-2 is 4.36 MPa, and the elongation is 1.8%. After three deep cryogenic treatments, the tensile strength of the CFs-6-3 is 4.41 MPa, and the elongation is 3.4%. Compared with CFs, after one, two, and three deep cryogenic treatments, the tensile strength increased by 2.4%, 8.1%, and 9.4%, respectively. The elongation rate increased by 1 time, 1.57 times, and 3.85 times, respectively. With the increase in cryogenic times, the strength of CFs improved to a certain extent. This might have been due to the fact that during the deep cryogenic treatment, the volume shrinkage effect caused the CFs to be compressed in the radial direction, resulting in the graphite layer being compacted and the amorphous region being reduced, thus improving the mechanical properties.

The electrochemical behavior test diagram of Pd@CFs-6-2 is shown in Figure 5. The CV diagram of the sample in the mixed solution of 1 M KOH and 1 M CH_3_OH at different sweeping speeds (Figure 5a), the peak current and voltage scatter diagram at different rates (Figure 5b), the relationship between the *j*_p_ and *v*^1/2^ diagram (Figure 5c), and the relationship between the *E*_p_ and *ln*(*v*) diagram (Figure 5d) were obtained. It can be seen from Figure 5a,b that as the scanning rate increases from 20 mV·s^−1^ to 200 mV·s^−1^, the sample peak current density *j*_p_ gradually increases. In contrast, the *E*_p_ corresponding to the peak current density gradually shifts positively. Figure 5c is obtained by fitting *j*_p_ and *v*^1/2^ corresponding to different sweeping speeds. It can be seen from the figure that *j*_p_ and *v*^1/2^ show a linear relationship of *y* = 495.4526*x* + 1863.56, *R*^2^ = 0.90668, indicating that the electrocatalytic oxidation of Pd@CFs-6-2 sample is a diffusion-controlled process. In addition, Figure 5d is obtained by fitting *E*_p_ and *ln*(*v*) corresponding to different sweeping speeds. From the figure, *E*_p_ and *ln*(*v*) show a linear relationship of *y* = 0.10213*x* − 0.3348 and *R*^2^ = 0.93943. This indicates that the electrocatalytic oxidation of methanol occurring on the surface of the Pd@CFs-6-2 samples is an irreversible process.

According to the experimental results above, it can be concluded that the deep cryogenic treatment has a positive effect on the Pd@CFs noble metal catalyst, and the catalytic performance is improved when the number of deep cryogenic treatments is two. To further investigate the effect of cryogenic time on the catalytic performance, the uncryogenic sample Pd@CFs was used as a control, and the catalyst samples were subjected to secondary deep cryogenic treatment with durations of 6 h, 12 h, 24 h, and 36 h.

The XRD of samples with different deep cryogenic treatment time and their correlation patterns of crystal plane texture coefficients are shown in Figure 6. From Figure 6a, it can be observed that the number and position of characteristic peaks of the samples treated at different deep cryogenic times have not changed, corresponding to the face-centered cubic structure of Pd. Further comparison shows that the intensity of characteristic peaks in the crystal plane of the samples is obviously enhanced after secondary deep cryogenic treatment at different times. With the extension of the deep cryogenic treatment time, the (*111*) and (*200*) crystal planes showed a trend of strengthening first and then weakening, among which the (*111*) and (*200*) crystal planes showed the highest intensity when the treatment time was 24 h. Figure 6b shows the trend chart of the texture coefficient of the Pd@CFs crystal plane at different deep cryogenic treatment time. It can be clearly seen that the orientation degree of (*200*) crystal plane of Pd@CFs samples with different deep cryogenic time is greatly improved, among which the preference degree of (*200*) crystal plane of the Pd@CFs-24-2 sample is the best, and the texture coefficient of (*111*) crystal plane is less than 1 due to the selection of standard baseline and the enhancement of diffraction peak of (*200*) crystal plane. Figure 6c shows the specific texture coefficient value of each crystal plane calculated according to the texture coefficient formula of the crystal plane. It can be seen that the Tc_(*200*)_ value of the Pd@CFs-24-2 is 1.183, which is 0.225 higher than that of 0.958 without deep cryogenic treatment. That is, after two deep cryogenic treatments for 24 h, the sample shows that the (*200*) crystal plane orientation degree is deepened, the (*220*) crystal plane texture coefficient is also partially improved, and there are also preferred orientations to varying degrees. The intensity of diffraction peaks increases in different degrees, mainly due to the crystal defects caused by the internal stress of metal during deep cryogenic treatment. In the deep cryogenic recovery stage, the metal grains rotate to a favorable position, and with the extension of deep cryogenic treatment time and cryogenic times, the internal stress increases cumulatively, thus strengthening the process. Similarly, from the changing trend of half the maximum width of (*111*) crystal plane, it can be observed that with the extension of deep cryogenic treatment time, the grains of samples gradually refine due to the cryogenic shrinkage effect. For the Pd@CFs-36-2, the intensity of its (*111*) crystal plane characteristic peak is weakened compared with that of the Pd@CFs-24-2 sample, which may be due to the long deep cryogenic treatment time and the stress inside the metal reaching a relatively balanced state, resulting in the weakening of its shrinkage effect compared with that of the Pd@CFs-24-2 sample, and then the intensity of crystal plane characteristic diffraction peak decreases.

The SEM images of samples with different deep cryogenic treatment time are shown in Figure 7. The SEM image of the Pd@CFs-12-2 sample at 10 μm magnification is shown in Figure 7a. It can be seen from the figure that the Pd is uniformly dispersed, but there is a small amount of agglomeration, and the fiber still maintains a spatial network structure. The SEM image of the Pd@CFs-24-2 sample at 10 μm magnification is shown in Figure 7b, the Pd loading state is more dispersed, the distribution is more uniform, and there is very little Pd agglomeration. This may be due to the extension of cryogenic time, the increase in the number of grain rotations, the decrease in the number of disordered particles stacked together, and the macroscopic loading state being more dispersed and more uniform, as well as the further decrease in the number of disordered particles of the Pd@CFs-24-2 sample, which can show that with the further extension of cryogenic time, the loading state of noble metals on the surface of CFs carrier is further improved. The SEM image of the Pd@CFs-36-2 sample at 10 μm magnification is shown in Figure 7c. From the diagram, it can be observed that the Pd loading on the surface of CFs has decreased, which may be due to the long deep cryogenic treatment time and the volume shrinkage effect of noble metal, which makes the atomic bonding too tight, which affects the adhesion with the CF support and causes the loss of noble metal. Based on the SEM pictures of each sample, the duration of deep cryogenic treatment has a certain influence on the loading state and loading capacity of precious metals. It is speculated that the duration of deep cryogenic treatment will cause changes in the electrocatalytic performance of the methanol of the catalyst sample.

The deep cryogenic treatment can improve the loading state of Pd@CFs-24-2 samples, so the Pd@CFs-24-2 samples were selected for the XPS test. The total XPS spectra of the Pd@CFs-24-2 samples are shown in Figure 7d after standard charge correction using C1s orbital 284.8 eV. From the figure, it can be observed that C, O, and Pd elements appear on the sample surface, and the 3d characteristic double peak of Pd appears in the range of 336 eV to 346 eV [29]. Figure 7e is the XPS spectrum of Pd 3d. Pd’s 3d characteristic double peaks are the Pd 3d5/2 characteristic peak at 335.64 eV and the Pd 3d3/2 characteristic peak at 340.98 eV, respectively, corresponding to the Pd^(0)^ valence state. After peak separation treatment, Pd 3d_5/2_ and Pd 3d_3/2_ peaks corresponding to the Pd^2+^ valence state appeared at 337.38 eV and 343.30 eV [30]. It can be seen that the secondary deep cryogenic treatment with a duration of 24 did not change the catalyst composition, the Pd in the sample still mainly exists in the metallic state, and a trace amount exists in the oxidized state [31]. Compared with the preparation of the Pd@CFs-6-2 sample, the binding energy is increased, which is usually accompanied by the downward shift in the d-band center, which can weaken the adsorption strength of Pd to the CO* intermediate [32], which is consistent with the negative shift in the peak voltage mentioned above.

The series of maps of the electrocatalytic oxidation methanol performance test of the samples with different cryogenic time are shown in Figure 8. The samples with different cryogenic treatment time were scanned in 1 M KOH solution at a scanning rate of 50 mV·s^−1^ in the potential range of −1~0.2 V (vs. SCE), and the CV curve was obtained as shown in Figure 8a. It can be observed from the diagram that there is no significant change in the CV curve profile of the sample after the second cryogenic treatment of different time lengths. It can be observed from the figure that there is no obvious change in the CV curve profile of the samples after secondary deep cryogenic treatment with different durations. It can also be found that the characteristic regions are obvious in different potential ranges, in which the hydrogen-related adsorption and desorption processes occur in the potential range of −1~−0.7 V, and the oxide formation process of Pd occurs in the potential range of 0~0.2 V. The negative scanning (−0.5~−0.2 V) potential region corresponds to the reduction characteristic peak of PdO. The EASA values obtained by calculating the reduction peak area of the CV curves of samples with different cryogenic time are shown in Figure 8b. From this it can be seen that the EASA values of Pd@CFs, Pd@CFs-6-2, Pd@CFs-12-2, Pd@CFs-24-2, Pd@CFs-36-2, and commercial Pd/C samples are 14.8 m^2^∙g^−1^, 23.3 m^2^∙g^−1^, 24.5 m^2^∙g^−1^, 26.8 m^2^∙g^−1^, 25.3 m^2^∙g^−1^, and 0.3 m^2^∙g^−1^, respectively. That is, with the increase in deep cryogenic treatment time, the EASA value of the sample showed a trend of first increasing and then decreasing. However, the EASA value of the sample after secondary deep cryogenic treatment of different durations was higher than that of the sample without deep cryogenic treatment. Among which the EASA value of the sample the Pd@CFs-24-2 after secondary deep cryogenic treatment time of 24 h was the largest, which was about 89.3 times that of commercial Pd/C.

The CV curves of samples with different cryogenic treatment time in 1 M KOH and 1 M CH_3_OH at a scan rate of 50 mV·s^−1^ are shown in Figure 8c. It can be seen from the figure that with the increase in deep cryogenic treatment time, the peak current density of forward scanning of samples in cyclic voltammetry test shows a trend of first increasing and then decreasing, and the peak current density of samples after deep cryogenic treatment is higher than that of samples without deep cryogenic treatment. The peak current density and voltage of the samples with different cryogenic times are shown in Figure 8d. It can be seen from the figure that when the sample is subjected to deep cryogenic treatment for 24 h, the electrocatalytic oxidation activity of Pd@CFs-24-2 sample for CH_3_OH is the largest, and the peak current density is 5220 A∙g^−1^, which is about 33% higher than that of the sample without deep cryogenic treatment (3931 A∙g^−1^) and 43.4 times that of the commercial Pd/C catalyst (120.4 A∙g^−1^). This shows that, based on secondary cryogenic treatment, the catalytic effect of the catalyst can be improved by prolonging the cryogenic treatment time. This is mainly due to the fact that the catalytic activity of the noble metal Pd is greatly affected by the crystal plane. Combined with the XRD analysis in Figure 2b, it can be seen that the Pd catalyst has a preferred orientation of the crystal plane during the secondary cryogenic treatment for different time periods. The crystal plane rotation changes the activity and selectivity of the Pd catalyst for the methanol catalytic reaction so that the catalytic performance is improved. Moreover, combined with the analysis of Figure 7b, it can be seen that cryogenic treatment can effectively improve the distribution of precious metals on the carrier so that the Pd catalyst exposes more active sites and the catalytic performance is improved. At the same time, further comparison showed that the oxidation potentials of Pd@CFs-6-2, Pd@CFs-12-2, Pd@CFs-24-2, and Pd@CFs-36-2 samples after deep cryogenic treatment were −0.027 V, −0.072 V, −0.039 V, −0.011 V, respectively. Compared with the Pd@CFs samples without deep cryogenic treatment (0.094 V), there was a negative shift in methanol oxidation potential, which indicated that deep cryogenic treatment could not only improve the electrocatalytic oxidation activity of the catalyst for CH_3_OH.

The i-t curves of samples with different cryogenic treatment times are shown in Figure 8e. At the beginning of the test, each sample has a high current density, but with the passage of time, the adsorption of intermediate products on the Pd surface hinders the contact between methanol and active sites, so the current density of the sample drops sharply. As the reaction continues, the current density of each sample tends to be steady. The steady-state current density curve of each sample at 5000 s is shown in Figure 8f. It can be seen from the figure that the steady-state current density of the sample shows a trend of first increasing and then decreasing with the increase in cryogenic duration. Among them, the steady-state current density of Pd@CFs-6-2, Pd@CFs-12-2, Pd@CFs-24-2, Pd@CFs-36-2 samples is 50.66 A∙g^−1^, 55.95 A∙g^−1^, 59.76 A∙g^−1^, and 52.79 A∙g^−1^. It is about 6.26 times that of commercial Pd/C (9.54 A∙g^−1^), which shows that increasing the deep cryogenic treatment time can improve the electrocatalytic stability of catalyst samples, and the deep cryogenic treatment time of 24 h has the best effect.

The CV curve of the Pd@CFs-24-2 sample in 1 M KOH and 1 M CH3OH mixed solution 500 times is shown in Figure 8g. It can be observed that with the increase in the number of scans, the peak current density shows a trend of first increasing and then decreasing. At the end of 500 cycles, the peak current density decreases from 5220 A∙g^−1^ in the first cycle to 5082 A∙g^−1^ in the 500th cycle, with an attenuation rate of 2.64%, which is much lower than the 12.14% attenuation rate of Pd@CFs-6-2 in the first experiment. Therefore, the cycle stability of the sample treated twice for 24 h is good, and the stability of its electrocatalytic oxidation of methanol is further improved. The current density changes in the Pd@CFs-24-2 at 100, 200, 300, 400, and 500 cycles are shown in Figure 8h. It can be seen from the figure that the peak current density of the sample does not change much after multiple cycles, which further proves that the stability of the sample is good.

In order to explore the influence of deep cryogenic treatment time on the mechanical properties of CFs carriers, for the convenience of description, CFs carriers without deep cryogenic treatment and with different deep cryogenic treatment times were named CFs, CFs-6-2, CFs-12-2, CFs-24-2, and CFs-36-2. The tensile properties of the carriers with different deep cryogenic treatment durations were tested, and the stress–strain curves of each sample were shown in Figure 8i. It can be observed from the figure that with the application of external force, the strain of the sample gradually increases until fracture occurs. For the sample without deep cryogenic treatment, the tensile strength of CFs is 4.03 MPa and the elongation is 0.7%. After 6 h of secondary deep cryogenic treatment, the tensile strength of the CFs-6-2 was 4.36 MPa and the elongation was 1.8%. After the secondary deep cryogenic treatment for 12 h, the tensile strength of the CFs-12-2 was 4.81 MPa and the elongation was 1%. After the secondary deep cryogenic treatment for 24 h, the tensile strength of the CFs-24-2 was 5.02 MPa and the elongation was 1.1%. After 36 h of secondary deep cryogenic treatment, the tensile strength of the CFs-36-2 was 4.51 MPa and the elongation was 1.2%. Compared with CFs, the tensile strength increased by 8.1%, 19.4%, 24.6%, and 11.9% after deep cryogenic treatment for 6 h, 12 h, 24 h, and 36 h, respectively. The elongation rate increased by 1.57 times, 0.57 times, 0.69 times, and 0.8 times, respectively. With the increase in the deep cryogenic treatment time, the tensile strength of the samples after secondary deep cryogenic treatment shows a trend of first increasing and then decreasing, and all of them are higher than those without deep cryogenic treatment. This may be because, with the increase in deep cryogenic treatment time, the volume shrinkage effect of the fiber matrix is enhanced. The CFs are further compacted by hoop stress in the radial direction, which makes the uneven layer structure of the fiber surface layer more compact, thus making it bear greater load in the radial direction and macroscopically showing that the tensile strength of the sample is improved. It can also be observed from the figure that when the deep cryogenic treatment time is increased to 12 h, 24 h, and 36 h, although the elongation of the sample is reduced compared with the sample with a 6 h treatment time, the tensile strength is slightly improved. The overall elongation and tensile strength are higher than those of the sample without deep cryogenic treatment; that is, the mechanical properties are improved. The reason for the decrease in analytical elongation may be that the deep cryogenic treatment time is too long, and the volume shrinkage effect of fiber reaches the limit, which makes the improvement effect of mechanical properties of the matrix also reach the limit.

## 4. Conclusions

Deep cryogenic treatment can promote the grain rotation of Pd and preferred crystal planes orientation, resulting in a uniformly distributed Pd load and improved catalyst the electrocatalytic activity and stability. Following deep cryogenic treatment refining, the catalytic activity of the Pd@CFs-24-2 catalyst is 5220 A∙g^−1^, which is 1.33 times that of the Pd@CFs and 43.4 times that of the commercial Pd/C catalyst, and the peak voltage of all catalysts after deep cryogenic treatment shifted negatively. The steady-state current density of Pd@CFs-24-2 reaches 59.76 A∙g^−1^, and its stability is 1.4 times that of Pd@CFs and 6.26 times that of commercial Pd/C catalysts. Regarding the mechanical characteristics of CF following deep cryogenic treatment, the tensile strength of CF increases in comparison to untreated CF. The catalyst’s tensile strength and the elongation rates increased by 24.6%, and 57%, respectively. Although the elongation is higher than that of samples without deep cryogenic treatment, it decreases when the cryogenic time is too long. Based on the above results, the deep cryogenic treatment can regulate the crystal plane orientation to increase the catalytic performance and reaction kinetics of the material, which is helpful to improve the efficiency of the catalyst for electrocatalytic oxidation of methanol, and the highly active the Pd@CFs-24-2 catalyst prepared by it has potential application prospects in DMFC.

## Figures and Tables

**Figure 1 nanomaterials-15-00338-f001:**
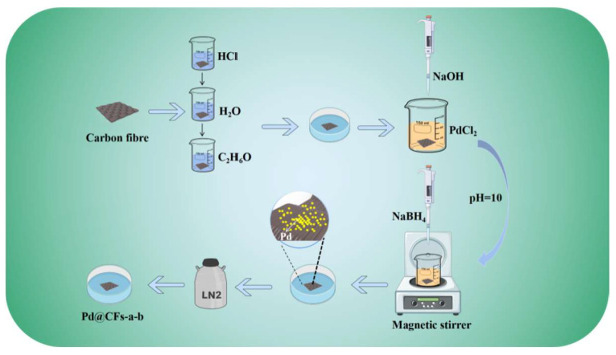
Flow chart of catalyst preparation.

**Figure 2 nanomaterials-15-00338-f002:**
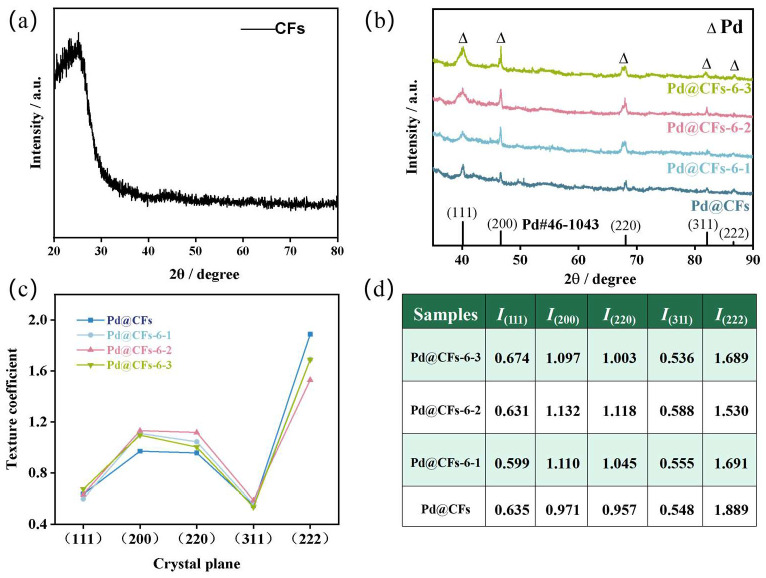
XRD pattern of CFs (**a**), XRD pattern of samples with different cryogenic times (**b**), change trend in crystal plane texture coefficient of samples with different cryogenic times (**c**), and texture coefficient values of each crystal plane of samples with different cryogenic times (**d**).

**Figure 3 nanomaterials-15-00338-f003:**
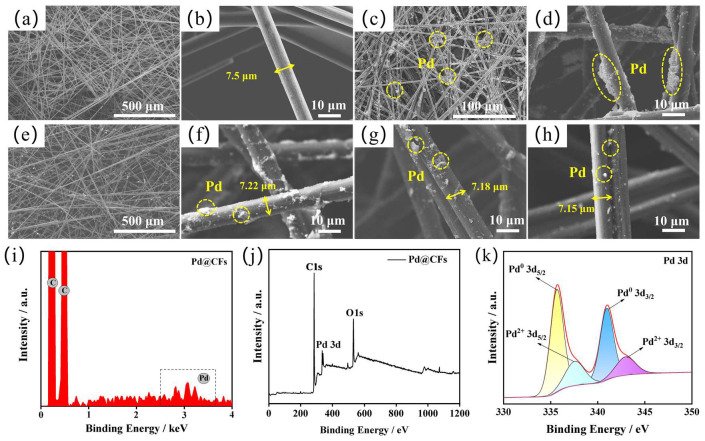
CFs (**a**,**b**), Pd@CFs (**c**,**d**), Pd@CFs-6-1 (**e**,**f**), Pd@CFs-6-2 (**g**), Pd@CFs-6-3 (**h**), EDS spectra (**i**), XPS spectra (**j**), and Pd 3d spectra (**k**) of Pd @ CFs.

**Figure 4 nanomaterials-15-00338-f004:**
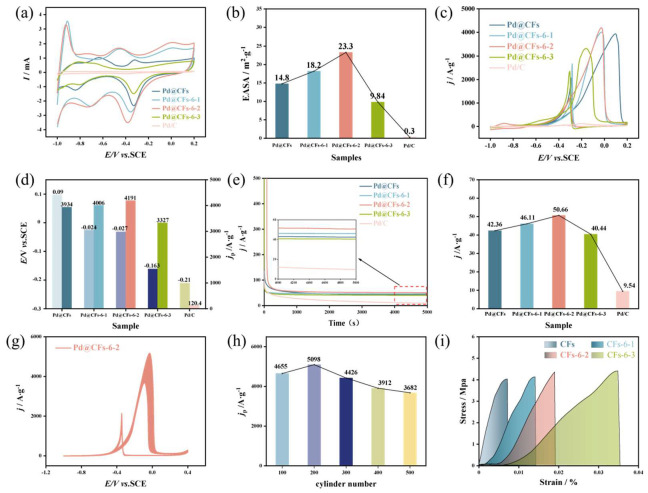
CV curves (**a**) and EASA values (**b**) of samples with different number of deep cryogenic treatments, CV curves in 1 M KOH and1 M CH_3_OH with a scan rate of 50 mV∙s^−1^ (**c**), peak current densities and voltages of samples with different number of deep cryogenic treatments (**d**), chronoamperometric current curves of samples with different number of deep cryogenic treatments (**e**), steady-state current densities at 5000 s (**f**), 500-cycle CV diagram of the Pd@CFs-6-2 sample (**g**), peak current density (**h**) for samples with different number of cycles, stress–strain curves for CFs, and samples with different number of deep cryogenic treatments (**i**).

**Figure 5 nanomaterials-15-00338-f005:**
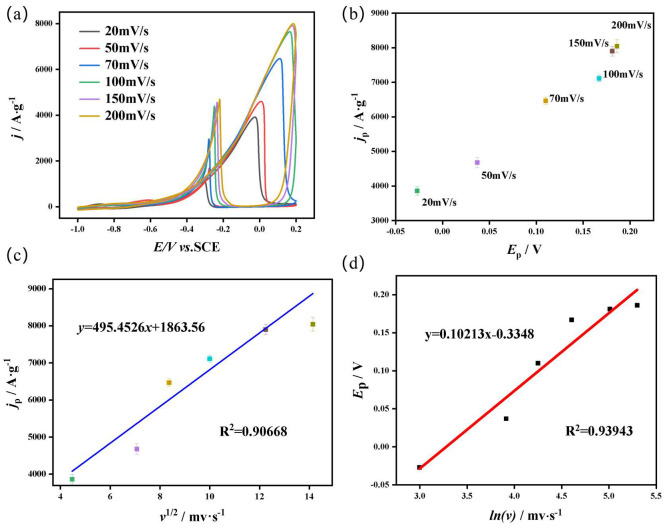
Pd@CFs-6-2 CV curves (**a**), *E*_p_ and *ln(v)* scatter plots (**b**), *j*_p_ vs. *v*^1/2^ relationship plot (**c**), *E*_p_ vs. *ln*(*v*) relationship plot (**d**) of samples in a mixed solution of 1 M CH_3_OH and 1 M KOH at different scanning speeds.

**Figure 6 nanomaterials-15-00338-f006:**
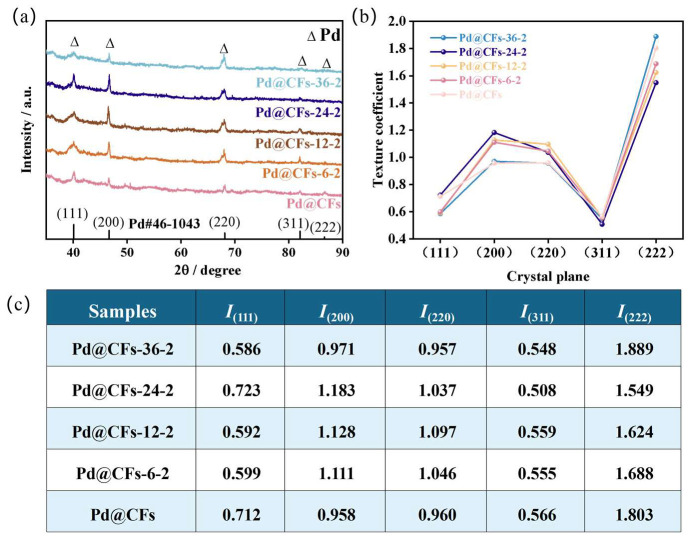
XRD patterns of Pd@CFs samples with different cryogenic times of deep cryogenic treatments (**a**), trend plot of Pd@CFs crystal surface texture coefficients at different deep cryogenic treatments times (**b**), and values of individual crystal surface texture coefficients at different deep cryogenic treatments times of Pd@CFs (**c**).

**Figure 7 nanomaterials-15-00338-f007:**
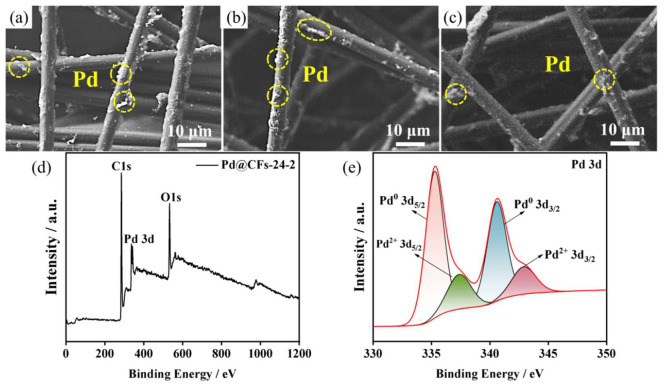
Pd@CFs-12-2 (**a**), Pd@CFs-24-2 (**b**), Pd@CFs-36-2 (**c**), XPS spectra (**d**), and Pd 3d spectra (**e**) of Pd@CFs-24-2.

**Figure 8 nanomaterials-15-00338-f008:**
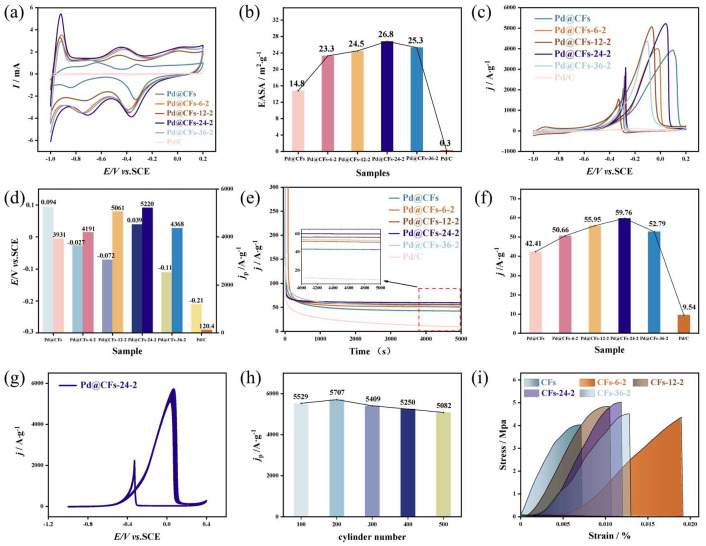
CV curves (**a**) and EASA values (**b**) of samples with different time of deep cryogenic treatments, CV curves in 1 M KOH and 1 M CH_3_OH with a scan rate of 50 mV∙s^−1^ (**c**), peak current densities and voltages of samples with different time of deep cryogenic treatments (**d**), chronoamperometric current curves of samples with different time of deep cryogenic treatments (**e**), steady-state current densities at 5000 s (**f**), 500-cycle CV diagram of the Pd@CFs-24-2 sample (**g**), and peak current density (**h**) for samples with different time of cycles, stress–strain curves for CFs, and samples with different time of deep cryogenic treatments (**i**).

## Data Availability

Data are contained within the article.

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
