# Peer review of "The Effect of Deep Cryogenic Treatment on the Electrocatalytic Performance of a Pd@CFs Catalyst for Methanol Oxidation"

_nanomaterials, 2025, doi:10.3390/nano15050338_

Round 1
Reviewer 1 Report
Comments and Suggestions for Authors
I think the paper deserves consideration. There are however several points which need to be improved before a positive recommendation is given.
1) Fig. 2. Why does the texture coefficient decrease after prolonged deep treatment ?
2) At lines 149-150 it is stated that:
"Generally speaking, the half-maximum width of diffraction peaks gradually widened with the increase of the deep cryogenic times. "
However inspection of Fig. 2 shows that the (111) and (220) widths have decreased with deep cold treatment. Please clarify.
3) Fig. 4h. What is the reason for the non monotonic dependence of the peak current density as a function of the number of cycles ?
4) In Fig. 5 the error bars on the current density must be provided. Without them it is impossible to determine whether the linear fit is acceptable or not. It seems indeed that a linear approximation is too rude to describe the velocity dependence shown in the figure and an S-shaped function seems at first more appropriate (at least if the error bars are definitely smaller than the difference between subsequent points).
5) Fig. 7 and related text: did you check with XPS whether Pd has decreased in the Pd@CFs-36-2 sample ?
Author Response
|
Comments 1: Fig. 2. Why does the texture coefficient decrease after prolonged deep treatment ? |
|
Response 1: Thank you for pointing this out.I would like to provide the following explanation for your question. The deep cryogenic treatment in a short period of time can cause rapid structural adjustment, such as the rearrangement of dislocations, which leads to the preferential formation of ( 200 ) and ( 220 ) crystal orientation grains, thereby increasing the texture coefficient. With the extension of the deep cryogenic treatment time, the recovery or recrystallization process occurs, the grains begin to rearrange, and the orientation tends to be random, which leads to the decrease of texture coefficient. Considering the residual stress inside the material, in the early stage of the deep cryogenic treatment, the residual stress promotes the formation of grains oriented to ( 200 ) and ( 220 ) crystal planes, but its long-term deep cryogenic treatment relaxes the stress, resulting in the destruction of orientation order and the decrease of texture coefficient. Thank you very much for your valuable comments, which have been explained and supplemented in the manuscript. For details, see lines 151 to 157 on page 5. |
|
Comments 2: At lines 149-150 it is stated that: "Generally speaking, the half-maximum width of diffraction peaks gradually widened with the increase of the deep cryogenic times. " However inspection of Fig. 2 shows that the (111) and (220) widths have decreased with deep cold treatment. Please clarify. |
|
Response 2: Thank you for your careful review of this article and professional correction of XRD pattern data. After verification, when we processed the XRD pattern of the Pd@CFs samples, we misplaced the first layout of the baseline data misplaced. We deeply apologize for this, and the text expression is expressed according to the correct map analysis. After receiving your opinion, we take measures immediately. All experimental map data have been comprehensively reviewed to ensure that the data correspond to the diagram one by one, and data verification links have been added to implement a two-person review system to prevent similar omissions. We attach great importance to the rigor of the research data. This omission has sounded the alarm for us, and the quality management process will be further improved in the future. Thank you again for your professional correction, which is of great significance to improve the quality of this study. Specific modifications are shown in Figure 2 ( b ) on page 4 and Figure 6 ( a ) on page 11 of the manuscript. |
|
Comments 3: Fig. 4h. What is the reason for the non monotonic dependence of the peak current density as a function of the number of cycles ? |
|
Response 3: Thank you for pointing this out.I would like to provide the following explanation for your question. In the initial stage of the CV test, the catalyst goes through the activation stage. In this process, the Pd active sites on the electrode surface are gradually activated, making the methanol oxidation reaction easier, resulting in an increase in peak current density. In addition, the CF carrier also has a certain adsorption and activation effect on methanol in the initial stage. The CF has a large specific surface area and good conductivity, which can provide good support for Pd and promote electron transport. At the beginning of the test, the interaction between CF and methanol may gradually increase, which further promotes the oxidation of methanol by Pd, resulting in a gradual increase in peak current density. However, in the process of methanol oxidation, intermediate products such as CO will be produced. These intermediates will adsorb on the surface of the catalyst and occupy the active sites, resulting in catalyst poisoning. As the number of CV tests increases, the intermediate products gradually increase, and the poisoning effect gradually increases, so the peak current density decreases. Therefore, the peak current density has a non-monotonic dependence on the number of cycles. We have added references for this, please refer to line 294 on page 8 for details. |
|
Comments 4: In Fig. 5 the error bars on the current density must be provided. Without them it is impossible to determine whether the linear fit is acceptable or not. It seems indeed that a linear approximation is too rude to describe the velocity dependence shown in the figure and an S-shaped function seems at first more appropriate (at least if the error bars are definitely smaller than the difference between subsequent points). |
|
Response 4: Thank you for your valuable suggestions on experimental data visualization. In response to the current density error analysis problem in Figure 5 that you pointed out, we have made the following improvements : In Figure 5 ( b ) on page 9 of the revision, we have added an error bar for all current density data points ( standard deviation of three independent experiments, n = 3 ). The updated data show that the relative standard deviation of each data point is less than the difference between each point, indicating that the experimental data has good reproducibility. The methodological suggestions proposed by you are very instructive, prompting us to conduct a more in-depth statistical demonstration of the choice of kinetic models. This significantly improves the scientific rigor of the research. Thank you ! |
|
Comments 5: Fig. 7 and related text: did you check with XPS whether Pd has decreased in the Pd@CFs-36-2 sample ? |
|
Response 5: Thanks to the reviewer 's advice, we are very sorry that we did not perform the relevant XPS test on the the Pd@CFs-36-2 sample. According to the existing research on the effect of cryogenic treatment on Pd, as shown in Synthesis and characterization of Pd doped MXene for hydrogen production from the hydrolysis of methylamine borane : Effect of deep cryogenic treatment. ( DOI : 10.1016 / J.JOEI.2023.101310 ). The Pd nanoparticles after cryogenic treatment are uniformly dispersed, and their XPS performance is similar to our test results.According to the CV test of the Pd@CFs-36-2 sample, it can be seen that the active area of the Pd@CFs-36-2 sample is small, and the peak current density has an obvious attenuation. Therefore, we speculate that the Pd loaded on the Pd@CFs-36-2 sample has a certain mass loss due to the long cryogenic treatment time. Thanks again to the reviewer 's comments, we have made corresponding modifications to the manuscript, as shown in 12 pages and 422 lines. |
Reviewer 2 Report
Comments and Suggestions for Authors
The study investigates the effects of deep cryogenic treatment (DCT) on the electrocatalytic performance of a flexible Pd@CFs catalyst for methanol oxidation. The key findings and implications suggest that DCT treated catalyst exhibit better catalytic activity. DCT caused Pd grains to rotate from disordered to ordered arrangements, leading to better dispersion of metal particles and exposure of more active sites. Tensile strength and elongation rates of the catalyst increased by 24.6% and 57%, respectively, compared to untreated CFs. If the following problems are well-addressed, this reviewer believes that the essential contribution of this article is vital for catalyst for methanol oxidation.
1. DCT treated catalyst only shows 1.33 times higher than Pd@CFs catalyst for methanol oxidation is it worthy to treat the catalyst at severely low temperatures for long duration. What is the motivation behind making DCT treated catalyst.
2. The high resolution XPS spectrum Pd in Fig 3 and Fig 7 (DCT treated 24 h 2 times) looks similar and there is no change in the oxidation states of Pd and electro catalysis is surface phenomena. However, authors did not observe any change in chemical state of Pd. How they justify the increase in electrocatalytic activity (1.33 times) of DCT treated catalyst.
3. Why authors did not run the stability test for long times such as 10 h or 20 h by comparing with commercial catalyst using chrono methods.
4. Authors need to explain the exact mechanism of electrocatalysis after DCT treatment and why it is showing superior performance with aid of any characterization techniques.
Author Response
|
Comments 1: DCT treated catalyst only shows 1.33 times higher than Pd@CFs catalyst for methanol oxidation is it worthy to treat the catalyst at severely low temperatures for long duration. What is the motivation behind making DCT treated catalyst. |
|
Response 1: Thank you for your attention to this technical route. The core motivation of our deep cryogenic treatment ( DCT ) research is not only to increase the peak current density, but also to significantly improve the reaction kinetic characteristics and practical application potential of the catalyst. Although the peak current density of the catalyst after DCT treatment is increased by 33 %, the peak potential of the sample after cryogenic treatment is negatively shifted compared with that before cryogenic treatment, which indicates that the activation energy barrier of the methanol oxidation reaction is significantly reduced. The catalyst can achieve high activity at a lower overpotential, which greatly improves the energy efficiency. This feature is particularly important for practical applications such as fuel cells. And the crystal plane preference of Pd nanoparticles can be regulated by simple DCT treatment. This optimization not only improves the initial activity, but also gives the catalyst better resistance to CO poisoning and stability. We agree that there is still room for improvement in the current activity increase, but this work reveals the kinetic regulation mechanism of DCT on precious metal catalysts for the first time, which provides a new idea for the development of low-cost and high-performance catalysts. Subsequent research will explore the structure-activity relationship between DCT parameters and performance in combination with your recommendations to achieve greater breakthroughs. The conclusion of the paper is modified, and the reaction kinetics of catalytic methanol promoted by cryogenic treatment is explained. Details can be found in lines 551, 552, 560 and 561 on page 15 of the manuscript. |
|
Comments 2: The high resolution XPS spectrum Pd in Fig 3 and Fig 7 (DCT treated 24 h 2 times) looks similar and there is no change in the oxidation states of Pd and electro catalysis is surface phenomena. However, authors did not observe any change in chemical state of Pd. How they justify the increase in electrocatalytic activity (1.33 times) of DCT treated catalyst. |
|
Response 2: Thank the reviewers for their important suggestions. About how XPS proves the relationship between the increase of electrocatalytic activity by 1.33 times, we make a detailed explanation here. The first is the explanation of the similarity of the spectra before and after XPS. The fact that the peak area before and after XPS does not change significantly indicates that cryogenic treatment has no effect on the loading state of Pd. Such as Synthesis and characterization of Pd doped MXene for hydrogen production from the hydrolysis of methylamine borane : Effect of cryogenic treatment. ( DOI : 10.1016 / J.JOEI.2023.101310 ).The binding energy of Pd0 3d5/2 in the Pd@CFs-6-2 sample is 335.28 eV, and the binding energy of Pd0 3d5/2 in the Pd@CFs-24-2 sample is 335.64 eV. The increase of binding energy is usually accompanied by the downward shift of d-band center ( Boosting fuel cell catalysis by surface doping of Pd nanocubes. DOI : 10.1016 / S1872-2067 (18 ) 63102-X ). It can weaken the adsorption strength of Pd to CO* intermediates, thus accelerating the methanol oxidation kinetics, which is consistent with the above peak potential negative shift results. I would like to thank the reviewers for their questions, and I would like to improve my manuscript. Please see lines 422 to 426 on page 12 for changes. |
|
Comments 3: Why authors did not run the stability test for long times such as 10 h or 20 h by comparing with commercial catalyst using chrono methods. |
|
Response 3: Thank the reviewers for their important suggestions. In terms of the choice of stability test time, we elaborate here : The core goal of this study is to explore the mechanism of the effect of cryogenic treatment on the initial activity and short-term stability of the Pd@CFs catalysts. The 5000 s chronoamperometry ( i-t ) test has been able to reflect the short-term decay behavior of the catalyst under typical operating conditions. Many similar studies usually use 1-5 hours of i-t test when initially verifying the stability of the catalyst. For example, the stability test of Pd/RGO and Pt/C catalysts in [1] Engineered Pt-Pd@RGO-KI nanosheet catalyst for enhanced methanol oxidation performance. ( DOI : 10.1016 / J.MCAT.2024.114771 ) was 1000 seconds. [2] Influence of different morphologies on the catalytic activity of Pt-Pd oxides for methanol oxidation. ( DOI : 10.1016 / J.ELECTACTA.2024.145241 ) The test time of PtPd/C catalyst is 1800 seconds. Our 5000 second test has covered this time range, and the data trend is consistent with the conclusion in the literature that ' stability decay mainly occurs in the first 1-2 hours '. The current study has preliminarily proved the effect of deep cryogenic treatment on the stability of the Pd@CFs catalyst through 5000 seconds test. Thank you for your valuable comments on this study, which is essential to improve our work. |
|
Comments 4: Authors need to explain the exact mechanism of electrocatalysis after DCT treatment and why it is showing superior performance with aid of any characterization techniques. |
|
Response 4: Thank you for your valuable suggestions on mechanism research. The mechanism of deep cryogenic treatment ( DCT ) to improve the performance of the Pd@CFs catalyst can be elucidated by the following characterization data : 1. XRD analysis : After DCT treatment, the (111) and (200) crystal plane texture coefficients of Pd are greater than 1 and the value is the largest, indicating that the crystal plane is preferentially oriented and crystal defects are generated. The metal grains rotate to a favorable position, which is conducive to grain refinement to improve the uniformity of the crystal structure and increase the active site of the sample for electrocatalytic oxidation of methanol. 2. XPS results : The Pd 3d binding energy of the sample after deep cryogenic treatment is positively shifted, which is usually accompanied by the downward shift of the d-band center, which can weaken the adsorption strength of Pd on CO* intermediates and enhance the stability of its electrocatalytic oxidation. 3. The calculation of CV active area : The ECSA value of the Pd@CFs-24-2 sample was 26.8 m2·g-1, while the ECSA value of the sample without deep cryogenic treatment was 14.8 m2·g-1, and the electrochemical active surface area of the catalyst was increased by 181%, confirming that more Pd active sites were exposed. 4. CV curve : The peak potential of the samples after deep cryogenic treatment all shifted negatively, indicating that the activation energy barrier of methanol oxidation reaction ( MOR ) decreased. 5. i-t test : The steady-state current density of the Pd@CFs-24-2 sample ( 59.76 A·g-1 ) was higher than that of the sample without deep cryogenic treatment ( 42.41 A·g-1 ), indicating that the defect engineering enhanced the binding force of Pd-support and reduced the stripping of Pd in the electrochemical cycle. DCT achieves the simultaneous improvement of activity and stability through the synergistic effect of lattice strain regulating electronic structure, Pd uniformity and stabilizing active sites, and optimizing the reaction kinetic path. |
Round 2
Reviewer 1 Report
Comments and Suggestions for Authors
The authors have considered all my comments.
I think that the error bars on the current density should be shown also in fig. 5c. The errors on the fitting parameters should be given and the number of digits in the fitting parameters should be revised accordingly.
Author Response
|
Comments : I think that the error bars on the current density should be shown also in fig. 5c. The errors on the fitting parameters should be given and the number of digits in the fitting parameters should be revised accordingly. |
|
Response : We sincerely appreciate the reviewer's valuable suggestions. In response to your professional comments, we have thoroughly revised the relevant figure content and related descriptions to ensure academic rigor. The specific modifications can be found in the text on page 9 (line 327) and the optimized Figure 5(c) of the manuscript. These revisions have significantly enhanced the scientific validity and accuracy of the presentation. We are truly grateful for your expert guidance in improving the quality of this work. |
Reviewer 2 Report
Comments and Suggestions for Authors
I am satisfied with the author's reply and accept the manuscript in its present form.
Author Response
Dear Reviewer :
Thank you for your valuable suggestions and hard work for this article! Your professional opinions make the research more rigorous, and detailed feedback adds a lot to the paper. I want to express my sincere respect to you. Your rigorous academic attitude and selfless spirit of guidance are imposing.